# Facile Synthesis of Di-Mannitol Adipate Ester-Based Zinc Metal Alkoxide as a Bi-Functional Additive for Poly(Vinyl Chloride)

**DOI:** 10.3390/polym11050813

**Published:** 2019-05-06

**Authors:** Yuepeng Li, Degang Li, Wenyuan Han, Manqi Zhang, Bing Ai, Lipeng Zhang, Hongqi Sun, Zhen Cui

**Affiliations:** 1School of Chemistry and Chemical Engineering, Shandong University of Technology, Zibo 255000, China; 13864306952@163.com (Y.L.); 18753375118@163.com (W.H.); 17853321663@163.com (M.Z.); hgxyaibing@163.com (B.A.); 2School of Engineering, Edith Cowan University, 270 Joondalup Drive, Joondalup, WA 6027, Australia; h.sun@ecu.edu.au; 3Weifang Zhengxuan Rare Earth Catalytic Materials Co., Ltd., Weifang 262737, China; zzciuzhen@sina.com

**Keywords:** PVC, thermal stabilizers, bi-functional, di-mannitol adipate ester-based zinc metal alkoxide, plasticizers

## Abstract

A new di-mannitol adipate ester-based zinc metal alkoxide (DMAE-Zn) was synthesized as a bi-functional poly(vinyl chloride) (PVC) thermal stabilizer for the first time. The materials were characterized with Fourier transform infrared spectroscopy (FT-IR) and thermogravimetric analysis (TGA). Characterization results confirmed the formation of Zn–O bonds in DMAE-Zn, and confirmed that DMAE-Zn had a high decomposition temperature and a low melting point. The thermal stability of DMAE-Zn on PVC also was tested by a conductivity test, a thermal aging test, and a UV-visible spectroscopy (UV-VIS) test. PVC stabilized by DMAE-Zn had a good initial color and excellent long-term stability. UV-VIS also showed that the conjugated structure in PVC stabilized by DMAE-Zn was almost all of the triene, suggesting that the addition of DMAE-Zn would suppress the formation of conjugated structures above tetraene. The dynamic processing performance of PVC samples tested by torque rheometer indicated that, having a good compatibility with PVC chains in the amorphous regions, DMAE-Zn contributed a good plasticizing effect to PVC. DMAE-Zn thus effectively demonstrates bi-functional roles, e.g., thermal stabilizers and plasticizers to PVC. Furthermore, FT-IR, a HCl absorption capacity test, and a complex ZnCl_2_ test were also used to verify the thermal stability mechanism of DMAE-Zn for PVC.

## 1. Introduction

Poly(vinyl chloride) (PVC) is one of the five general plastics, and has been widely used in infrastructure construction and home supplies because of its excellent low cost, high strength, corrosion resistance, and self-extinguishing capability, etc. [1]. In applications, its poor thermal stability is the biggest weakness of PVC. When the temperature reaches 130 °C or higher, PVC starts to break down and release hydrogen chloride (HCl), which can further aggravate this degradation process [2]. The color of PVC changes significantly from white to brown, and finally black, while its mechanical properties also decline rapidly [3]. Therefore, thermal stabilizers must be added during PVC processing to inhibit the production of HCl and remove the free HCl [4].

At present, lead salts, metal soap, and organotin are the mainstream thermal stabilizers used in PVC. Although lead salts provide the best thermal stability of PVC, their toxicity restricts their application. Metal soap, such as calcium stearate (CaSt_2_) and zinc stearate (ZnSt_2_), has no toxicity, but its thermal stabilizing effect on PVC is not ideal, especially as “zinc-burning” leads to the rapid degradation of PVC [5,6,7,8,9,10]. Organotin provides an excellent thermal stability of PVC, but its cost is too high [11]. It is then necessary to develop new non-toxic and efficient thermal stabilizers of PVC [12,13,14].

A number of new types of thermal stabilizers, for example, zinc and calcium salts of 11-maleimideoundecanoic acid, synthesized liquid Ca/Zn thermal stabilizer, zinc and calcium oxolinic complexes, ricinoleic acid-based Ca/Zn stabilizer have been reported [11,13,15,16,17,18,19]. Some researchers also indicated that polyols can serve as auxiliary thermal stabilizers. As they have many hydroxyl groups, polyols are able to chelate ZnCl_2_ to inhibit the “zinc-burning” phenomenon [3,20]. Jenneskens et al. found that natural polyols can significantly improve the thermal stability of PVC [21,22]. Guo et al. reported that the addition of pentaerythritol combined with CaSt_2_/ZnSt_2_ could improve the thermal and color stability of PVC [23].

Recently, we synthesized a series of polyol-based metal alkoxides as PVC thermal stabilizers, and showed that they were efficient in inhibiting the degradation of PVC [24,25]. There is still a challenge for these polyol-based metal alkoxides to be used as PVC thermal stabilizers, because their melting point (about 220 °C) [1,2] is significantly higher than the processing temperature of PVC, affecting their consistency with PVC [26]. In order to reduce the melting point of polyol-based metal alkoxides, Shentu et al. employed mannitol with a low melting point of 166 °C to synthesize mannitol-zinc metal alkoxides, which was then used to improve PVC thermal stability [2]. A new strategy was applied by our group to overcome this shortcoming by esterification. For example, we synthesized pentaerythritol stearate ester-based zinc alkoxides (PSE-Zn) [26] and cis-1,2-cyclohexanedicarboxylic acid di-mannitol ester-based zinc metal alkoxides [27]. We observed that the new ester-based alkoxides have a lower melting point and a good thermal stabilization performance on PVC.

Generally speaking, a plasticizer, one of the most important PVC processing aids, should be added to adjust the mechanical and thermal properties of PVC. Phthalate esters are the most commonly used plasticizers in PVC processing. However, most phthalate plasticizers are known to be toxic [28]. More and more researchers are committed to developing new non-toxic PVC plasticizers. It is recommended to use sustainable alternatives instead of phthalate esters. Epoxidized vegetable oils are one of these alternatives [28,29].

In this study, in order to lower the melting point of metal alkoxides, adipic acid was used to synthesize a new kind of thermal stabilizer, di-mannitol adipate ester-based zinc metal alkoxide (DMAE-Zn). Adipic acid was chosen because it has good compatibility with PVC, enabling the plasticizing effect of DMAE-Zn on PVC. The synthesized DMAE-Zn was characterized by FT-IR spectroscopy and thermogravimetric analysis. Their thermal stability performances of PVC also were tested through the oven aging method, a conductivity test, a UV-VIS spectroscopy test, and a torque rheometer test. It was found that DMAE-Zn can effectively have a bi-functional role in PVC, acting as both a thermal stabilizer and a plasticizer.

## 2. Experimental

### 2.1. Materials

PVC resin (type SG-5, average degree of polymerization: 1005, viscosity number: 67) was from Petrochemical Qilu Limited Co., Zibo, China. Some additives, such as lead salt stabilizers (the mixture of tribasic lead sulfate, dibasic lead phosphite, polyethylene wax, and assistant agents, PbO content: 30 ± 2%), CaSt_2_ (98%), ZnSt_2_ (98%), dioctyl phthalate (DOP, 99%), TiO_2_ (anatase titanium dioxide, 99%), chlorinated polyethylene (CPE, chlorine content: 35 ± 2%), acrylics copolymer (ACR, 99%), and CaCO_3_ (light calcium carbonate, 1800 mesh), were all of industrial grade and were kindly supplied by Shandong Huike Additives Co., Zibo, China. D-mannitol (≥99.0%), adipic acid (≥98%), cyclohexane (≥99.0%), zinc acetate (98%), and other chemical agents were all of analytical grade and were supplied by Shanghai McLean Biochemical Technology Co., LTD, Shanghai, China.

### 2.2. Preparation Process

#### 2.2.1. Preparation of Di-Maltitol Adipate Ester

Di-mannitol adipate ester (DMAE) was prepared by a traditional method. D-mannitol and adipic acid in a molar ratio of 2:1 were added into a mixer set and stirred for 5 min. Then, the mixture was put into a 500 mL three-necked round bottom flask with mechanical stirring and a reflux water device, followed by the introduction of cyclohexane as the dehydrant. The mixture was heated to 160 ± 1 °C for 3 h, to remove the excess cyclohexane with a vacuum pump. The DMAE was then obtained. 

#### 2.2.2. Preparation of Di-Mannitol Adipate Ester-Based Zinc Metal Alkoxide (DMAE-Zn)

DMAE-Zn was synthesized through the method of alcoholysis. Zinc acetate and DMAE in a molar ratio of 1:1 were mixed in a three-necked flask with a mechanical stirring. Excess absolute ethanol was added and the reactor was then heated to 160 ± 1 °C for 4 h. After the solvent was evaporated, the samples were washed with ethanol three times to obtain the product of DMAE-Zn.

### 2.3. Material Characterization

Samples were characterized by Fourier transform infrared (FT-IR) spectroscopy (Nicolet 5700, Beijing, China) with the KBr disc method. The spectral range was 400–4000 cm^−1^, with a scan rate 128 min^−1^ and a resolution of 4 cm^−1^. The blank spectrum was tested before each measurement to eliminate the spectrum subtraction resulting from carbon dioxide and water in the air. Thermogravimetric analysis (TGA) and derivative thermogravimetry (DTG) of the samples were performed using a Q500 analyzer (TA instruments, Selb, Germany) at a 10 °C min^−1^ heating rate from 25 to 700 °C in a nitrogen atmosphere. The results were used to investigate the decomposition of the materials [26]. 

### 2.4. PVC Sample Preparation

About 100.0 g of PVC, 5 mL of dioctyl phthalate (DOP), 4.0 g of TiO_2_, 9.0 g of chlorinated polyethylene (CPE), 2.0 g of acrylics copolymer (ACR), 20.0 g of CaCO_3_, 1.6 g of stearic acid, and 4 g of thermal stabilizers were added into a blender to undergo sufficient mixing. Then the mixture was milled using an open twin-roller (PX-GY-150, Shenzhen Pengxiang Yunda Machinery Technology Co., Shenzhen, China) at 180 °C for 5 min to produce the PVC sheets with a thickness of about 1.0 mm.

### 2.5. Thermal Stability Test

#### 2.5.1. Conductivity Measurement

PVC sheets were cut into small square pieces of 0.5 mm × 0.5 mm with a total weight of 2.0 g, and put into a home-made reaction vessel. The vessel was placed in an oil bath and heated to 180 °C. Then 60 mL of deionized water was added in a beaker of 100 mL to test its conductivity. The HCl gas produced during the thermal decomposition of PVC was introduced into the measuring beaker by the nitrogen stream (about 7 L/h). The HCl was absorbed by the deionized water in the measuring beaker so that the conductivity of water gradually changed with time. Therefore, the decomposition rate of PVC could be recorded through measuring the change of the conductivity of the deionized water. A conductivity meter (DDS-307, Shanghai Instrument Electric Scientific Instrument Co., Ltd., Shanghai, China) was used to measure the conductivity.

#### 2.5.2. Thermal Aging Test

PVC sheets were cut into small samples of 15 mm × 15 mm. These sheets were heated to 180 °C in an oven. The sheets were taken out after every 10 min. The thermal stability of PVC was evaluated by monitoring the color change.

#### 2.5.3. UV-VIS Spectroscopy Test

It is known that PVC begins to decompose and produce conjugated double bonds when the temperature reaches 130 °C. PVC samples were first dissolved by tetrahydrofuran (THF). Then the concentration of conjugated double bonds was measured by UV-visible spectrometer (UV-VIS). The UV-VIS spectra were recorded at 25 ± 5 °C using a UV-2450PC spectrometer with the slit width set at 2 nm over the wavelength range of 200–500 nm.

#### 2.5.4. Torque Rheometer Test

The effect of DMAE-Zn on the dynamic rheological properties of PVC was tested by a torque rheometer (RM-200C, Harbin Harp Electric Technology Co., Harbin, China). The operating temperature was set at 180 °C while keeping the rotor at a speed of 35 rpm. 

#### 2.5.5. Capacity for Neutralizing HCl

The capacity of stabilizers to neutralize HCl was investigated by conductometric titration experiments. First, 6.00 mL of 0.1 M HCl standard solution was diluted with 20.00 mL of ethanol and 10.00 mL of deionized water. Then, 0.0500 g of thermal stabilizer, such as DMAE-Zn, lead salts, ZnSt_2_, CaSt_2_, and so on was dissolved in this solution with a magnetic stirring at 40 °C. The excess HCl was back-titrated with 6.00 mL of 0.1 M NaOH standard solution. The conductivity of the solution was measured by a conductivity meter (DDS-307, same as in Section 2.5.1). The volume of NaOH solution corresponding to the minimum conductivity of the solution was the titration endpoint, and the capacity for neutralizing HCl was calculated by the volume of NaOH solution used.

## 3. Results and Discussion

### 3.1. Characterization of Metal Alkoxides

#### 3.1.1. Fourier Transform Infrared Spectroscopy

Figure 1 is the FT-IR spectra of the synthesized DMAE and DMAE-Zn. Figure 1 shows that both the DMAE and DMAE-Zn have strong peaks at 3400–3500 cm^−1^, which represent the characteristic absorbance band of –OH, as a result of the stretching vibration. The peak at about 2900 cm^−1^ corresponds to the stretching vibration of the –C–H groups. The absorption peak at about 1738 cm^−1^ can be assigned to the ester carbonyl bonds (–C=O). There were no absorption peaks at 1786 cm^−1^ of carboxylic acid carbonyl bonds in DMAE and DMAE-Zn, indicating that the esterification of stearic acid with D-mannitol was complete. The absorption bands between 650 and 550 cm^−1^ in Figure 1b are attributed to the –O–Zn bonds [24]. The appearance of –O–Zn bands indicated that the synthesized compound contained metal alkoxides.

There are six hydroxyl groups in a D-mannitol molecule. All of these hydroxyl groups can be esterified using adipic acid. Therefore, DMAE may contain more than 12 esters. Its derivative, DMAE-Zn, is expected to have a more complex structure. Scheme 1 shows a simplified reaction pathway for the synthesis of DMAE-Zn. Two carboxyl groups of adipic acid react with two alcoholic hydroxy groups of the di-mannitol to form esters. Then, through an alcoholysis reaction, the ethyoxyl of ethanol zinc will exchange with the hydroxy of polyol ester. After completely evaporating the ethanol, the targeted DMAE-Zn can be obtained.

#### 3.1.2. Thermogravimetric Analysis

Figure 2 shows the TGA, differential thermal analysis (DTA), and DTG curves of the targeted DMAE-Zn. There was one weight-loss step within the temperature range from 200 to 500 °C in the TG curve of the DMAE-Zn, which was accompanied by a maximum weight-loss peak. The DTG curve of Figure 2 shows that there are two pyrolysis temperatures for DMAE-Zn—265.8 and 342.5 °C. All the pyrolytic temperatures were far more than 180 °C, which is the processing temperature of PVC, meaning that the synthesized DMAE-Zn did not break down during PVC processing and was suitable for use as a thermal stabilizer of PVC. The TG curve in Figure 2 shows that the residue was 20.57%, which is slightly higher than the result of the elemental analysis. The residue might be composed of ZnO and carbon. The DTA curve in Figure 2 shows that the melting point of the targeted DMAE-Zn can be estimated from the first endothermic peak where the weight changes slightly. Thus, the melting point of DMAE-Zn can be estimated to be 154.4 °C, lower than the PVC processing temperature (180 °C). Therefore, the targeted metal alkoxides had good compatibility with PVC.

### 3.2. Thermal Stability Tests of DMAE-Zn on PVC

#### 3.2.1. Conductivity Test

In the tests, nitrogen gas, as a carrier gas, carries HCl gas released during the degradation of PVC into deionized water. The dissolution of HCl into the deionized water changes the conductivity of the water. As a result, the degradation rate of PVC can be estimated by testing the rate of change in the conductivity of the water. In the conductivity vs. time plot, the introduction time (*T*_i_) refers to the time from the start of the heating to when the conductivity starts to rise, while the stability time (*T*_s_) refers to the time where the conductivity of the water reaches 50 μS cm^−1^ [30], which can be considered as the maximum acceptable conductivity value in PVC degradation.

Figure 3 shows the conductivity vs. time plots for pure PVC and PVC stabilized by 4 phr (parts by weight per hundred parts of resin) of ZnSt_2_/CaSt_2_ (1:1), and DMAE-Zn. Generally speaking, the shorter the *T*_i_, the worse the color stability of PVC. It can be seen from Figure 3 that the *T*_i_ and *T*_s_ of pure PVC are 9.6 min and 21.8 min, respectively, indicating that the initial color stability of pure PVC is quite short and, after having been heated for 21.8 min, pure PVC might degrade completely. As for the PVC samples stabilized by ZnSt_2_/CaSt_2_, Figure 3b shows that the *T*_i_ and *T*_s_ of PVC stabilized by ZnSt_2_/CaSt_2_ are 16.1 min and 24.3 min, respectively. By comparing Figure 3a,b, although the *T*_i_ of PVC stabilized by ZnSt_2_/CaSt_2_ is longer than that of pure PVC, the *T*_s_ values of these two PVC samples are relatively close. This means that the initial stability of PVC stabilized with ZnSt_2_/CaSt_2_ was better than that of pure PVC. However, the long-term stability of PVC stabilized with ZnSt_2_/CaSt_2_ was worse. This is because ZnCl_2_ would be produced when ZnSt_2_ absorbed the HCl produced during PVC degradation. ZnCl_2_ was a strong Lewis acid, which would catalyze the degradation of PVC. When the amount of ZnCl_2_ accumulated to a certain extent, it would suddenly accelerate the degradation of PVC, leading to the rapid aging of PVC. This phenomenon is called “zinc-burning” [31]. It is worth noting that the *T*_i_ and *T*_s_ of PVC stabilized by DMAE-Zn were extended to 31.1 min and 67.9 min, respectively, indicating that this PVC sample had excellent color stability and long-term stability. This might be due to the fact that DMAE-Zn has many hydroxyl groups, which might complex with the produced ZnCl_2_ to inhibit the catalytic degradation of PVC.

#### 3.2.2. The Thermal Aging Test

When the temperature exceeds 120 °C, PVC starts thermal degradation. The unstable chlorine atoms, such as allyl chloride and tert-butyl chloride, combine with hydrogen atoms in the ortho to form hydrogen chloride, and the –C=C– comes into being. Further thermal degradation would produce a double bond of the conjugated structure. When the conjugated structure increases to a certain length, the color of PVC changes to light brown, then dark brown, and finally black.

Figure 4 shows the thermal aging test results of pure PVC and PVC with 4 phr of ZnSt_2_/CaSt_2_ (1:3), and 1 to 4 phr of DMAE-Zn. The initial color (heated for 0 min) of pure PVC was light yellowish gray, indicating that slight thermal degradation occurred during the preparation of pure PVC sheets with the two-roll tablet press. As the heating time increased, the color of pure PVC gradually changed into light brown (after 10 min), brown (after 30 min), and black (after 90 min). PVC samples stabilized by ZnSt_2_/CaSt_2_ (1:3) had an excellent initial color, showing that the formula of ZnSt_2_/CaSt_2_ can improve the initial color of PVC. It is worth noting from Figure 4 that, after having been heated for 40 min, the color of PVC stabilized by ZnSt_2_/CaSt_2_ turned dark brown quickly, indicating the “zinc-burning” phenomenon and a poor long-term thermal stability of the formula of ZnSt_2_/CaSt_2_ for PVC.

It can be seen from Figure 4 that the initial color of PVC sheets with 1 phr of DMAE-Zn was white and changed slowly into pale yellow brown and brown within 40 min, indicating that the addition of DMAE-Zn can improve the color stability of PVC. However, from 50 min, the color of the PVC stabilized by 1 phr of DMAE-Zn turned dark brown quickly within 10 min, showing that 1 phr of DMAE-Zn was not enough to improve the long-term thermal stability of PVC. Figure 4 proves that as the dosage of DMAE-Zn increases, the color of PVC can be improved significantly. It can also be seen from Figure 4 that the thermal stability of PVC stabilized by 2 phr and 3 phr of DMAE-Zn had a similar efficiency. PVC stabilized with 4 phr of DMAE-Zn had the best initial color, and the color started to slowly turn yellow from 60 min and did not turn completely back within 120 min. Therefore, 4 phr of DMAE-Zn had the highest thermal stabilizing efficiency.

#### 3.2.3. UV-VIS Spectroscopy Test

When the PVC is thermally degraded, a conjugated double bond structure is produced. The conjugated double bond structure has a distinct absorption peak in the UV-VIS spectrum. Furthermore, the length of the conjugated double bond affects the position of the absorption peak, and the greater the concentration of the double bond (C_db_), the higher the absorption peak. Figure 5 reveals the UV-VIS spectra of pure PVC and PVC stabilized by 4 phr of different thermal stabilizers which were heated for 0 min (A) and 60 min (B) at 180 °C. The maximum absorption peak of three samples was at about 280 nm, belonging to the conjugated triene structure formed in the dehydrochlorination of PVC. Pure PVC exhibited the highest C_db_, indicating that CaSt_2_/ZnSt_2_ and the targeted metal alkoxides were effective in improving the initial thermal stability of PVC. Between the two thermal stabilizers, the peak height of PVC stabilized with CaSt_2_/ZnSt_2_ was the higher one, suggesting that the formula with a ratio of CaSt_2_ to ZnSt_2_ of 3 was not effective in improving the initial color of PVC, probably because there was less ZnSt_2_ and more CaSt_2_ (ZnSt_2_ is the best additive in improving the initial color of PVC). As for the targeted DMAE-Zn, Figure 5A shows that the peak height of PVC stabilized by DMAE-Zn was the lowest, indicating the lowest C_db_ in PVC stabilized by DMAE-Zn. These results confirmed that DMAE-Zn is the most efficient in preventing PVC molecules from forming conjugated structures, and thus preventing the thermal degradation of PVC, consistent with the results of the oven thermal aging tests.

Compared with Figure 5A, the C_db_ of PVC samples in Figure 5B increased greatly. In particular, after having been heated for 60 min, the C_db_ of PVC stabilized with CaSt_2_/ZnSt_2_ having the largest value, even exceeding that of pure PVC. This is because the “zinc-burning” phenomenon occurred in PVC stabilized by CaSt_2_/ZnSt_2_. The results of the oven thermal aging tests (Figure 4) also confirmed this conclusion, as the color of the PVC sheet stabilized by CaSt_2_/ZnSt_2_ became dark gray after being heated for only 60 min. The C_db_ of PVC stabilized by the synthesized DMAE-Zn did not increase significantly, but only increased from 0.26 to 0.32, indicating that DMAE-Zn had the highest efficiency in suppressing the production of conjugated double bonds.

The position of absorption peaks should also be noted. Comparing Figure 5A,B, it can be seen that the position of these main absorption peaks did not shift, showing that most of the PVC thermal degradation products have a conjugated triene structure. However, the absorbance curve of pure PVC (see curve a in Figure 5A) showed that there was weak absorption at the range of 300–330 nm which belongs to tetraenes and pentaene. PVC samples stabilized by the two thermal stabilizers had no absorption at the same range. Figure 5B shows that, after having been heated for 60 min, the absorbance curve of pure PVC increased obviously in the range of 300–330 nm. This suggests that the C_db_ of tetraenes and pentaene in pure PVC had further increased. The absorbance curve of PVC stabilized by DMAE-Zn increased slightly in the range of 300–330 nm. Surprisingly, the absorbance curve of PVC stabilized by CaSt_2_/ZnSt_2_ still did not increase, indicating that the ZnCl_2_ produced from the reaction between ZnSt_2_ and HCl intended to catalyze the PVC degradation only formed a conjugated triene structure.

#### 3.2.4. Synergy between DMAE-Zn and CaSt_2_ or ZnSt_2_ on PVC Thermal Stability

The above results (Section 3.2.1, Section 3.2.2 and Section 3.2.3) indicate that DMAE-Zn is an efficient thermal stabilizer for PVC. In order to improve PVC color thermal stability and reduce manufacturing costs (if this method can be industrialized), the synergistic effect of DMAE-Zn with CaSt_2_ and ZnSt_2_ on PVC thermal stability was investigated. Figure 6 shows the oven thermal aging test results of PVC stabilized by CaSt_2_/DMAE-Zn and that stabilized by ZnSt_2_/DMAE-Zn. It can be seen from the figure that the color of PVC stabilized by 4 phr of CaSt_2_ started to turn light brown after being heated for 10 min, and turned darker brown at 60 min, showing that pure CaSt_2_ cannot improve the PVC thermal stability significantly. The color of PVC with 3 phr of CaSt_2_ + 1 phr of DMAE-Zn turned black at 60 min, indicating that the thermal stability of this formula is similar to that of pure CaSt_2_. With increasing DMAE-Zn, the thermal stability of PVC was improved quickly. For example, the color of PVC stabilized by CaSt_2_/DMAE-Zn with mass ratios of 2:2 and 1:3 did not turn black until 120 min. Moreover, the color of PVC stabilized by CaSt_2_/DMAE-Zn with a mass ration of 1:3 did not change within 30 min, and only turned slightly brown within the next 40 min, indicating that there existed a good synergistic effect between CaSt_2_ and DMAE-Zn on PVC thermal stability.

As for the system of ZnSt_2_/DMAE-Zn, Figure 6 shows that, when ZnSt_2_ was used alone as a PVC thermal stabilizer, PVC samples turned black quickly within 10 min due to the “zinc-burning” phenomenon. With the dosage of DMAE-Zn increasing, the initial color and long-term stability of PVC can be improved obviously. For example, the color of PVC stabilized by ZnSt_2_/DMAE-Zn with a mass ratio of 1:3 remained unchanged in 70 min, and did not turn black in 120 min. Therefore, there existed an excellent synergistic effect between ZnSt_2_ and DMAE-Zn on PVC thermal stability. This was probably due to the large number of hydroxyl groups in DMAE-Zn, which could complex with ZnCl_2_ to avoid catalyzing PVC thermal degradation.

#### 3.2.5. Torque Rheology Test of DMAE-Zn

The torque rheology test was conducted according to ASTM D 2538-02 [32], and operated at 180 °C as was described in Section 2.5.4. Figure 7 shows the results for pure PVC and PVC stabilized by DMAE-Zn. Both samples had two distinct peaks which were the feed peak and the plasticized peak of PVC resin. The first peaks for both samples appeared at almost the same time. The first peak height of PVC stabilized by DMAE-Zn was 50 Nm higher than that of pure PVC, indicating that the addition of DMAE-Zn did not improve the lubricating property of PVC. The second peaks of pure PVC and PVC stabilized by DMAE-Zn appeared at different time points. Pure PVC required approximately 111 s to complete the plasticization. In PVC stabilized by DMAE-Zn case, however, plasticization was completed in 55 s. The significantly reduced plasticizing time obtained by incorporating DMAE-Zn as compared to pure PVC confirmed the excellent plasticizing effect of DMAE-Zn on PVC. It is well known that plasticizers must have a good compatibility with PVC. For example, Gilbert et al. reported that plasticizers acted as solvents for amorphous regions of PVC, and the PVC chains in the amorphous regions might become solvated at elevated temperatures during processing [33]. Therefore, we concluded that DMAE-Zn had a good plasticizing effect on PVC because DMAE-Zn has a good compatibility with PVC chains in the amorphous regions. It is also worth noting that the balance torque of PVC stabilized by DMAE-Zn was slightly lower than that of pure PVC. Considering that DMAE-Zn had no lubricity to PVC, the lower balance torque of PVC stabilized by DMAE-Zn indicated that the addition of DMAE-Zn could decrease friction between PVC molecules through solvation.

### 3.3. The Thermal Stabilizing Mechanism of DMAE-Zn

In our previous work we found that, due to its high electronegativity, the alkyl oxygen of metal alkoxides had a tendency to attack the carbon atoms (given its high positive charge) attached to allyl chloride in PVC chains [34]. At the same time, allyl chloride with a high electronegativity would attack the zinc atom (having a high positive charge) of the metal alkoxides. Scheme 2 illustrates the reaction mechanism. This reaction mechanism also applies to DMAE-Zn since it contains metal alkoxides, which could explain the fact that PVC stabilized by DMAE-Zn had a better initial color (i.e., white) than pure PVC (refer to Figure 4). The allyl chloride in PVC formed a bond with the Zn atom in DMAE-Zn. The released Cl atoms from PVC were taken up by zinc ions in DMAE-Zn (Scheme 2a) to generate compound I, which prevented the formation of free ZnCl_2_. This explains the excellent long-term thermal stability observed for PVC stabilized by DMAE-Zn (Figure 4).

In order to verify this mechanism of DMAE-Zn, FT-IR spectra were used to test pure PVC sheets and PVC sheets stabilized by DMAE-Zn and heated at 180 °C for 0 min and 30 min. Figure 8 shows the FT-IR results. The obvious absorption peaks at 1096 cm^−1^ in curves c and d are the characteristic peaks corresponding to –C–O–C–. The existence of –C–O–C– improved the feasibility of the mechanism of DMAE-Zn to replace allyl chloride in PVC molecules—the alkyl oxygen of metal alkoxides attacked the carbon atom attached to allyl chloride in the PVC molecule and the allyl chloride atom attacked the zinc atom in DMAE-Zn (as shown in Scheme 2a), and then formed compound I. As a comparison, there was no characteristic absorption peak at 1096 cm^−1^ in curves a and b.

A conductivity titration experiment was carried out to verify the ability of DMAE-Zn to absorb HCl. The experimental method is described in Section 2.5.5. The results are shown in Table 1. Table 1 shows that lead salts had the highest capacity to absorb HCl, reaching 281.1 mg/g. The HCl absorption capacity of DMAE-Zn was 131.4 mg/g, which was larger than that of ZnSt_2_ and CaSt_2_, showing that DMAE-Zn could improve the long-term thermal stability of PVC.

In general, the addition of an auxiliary thermal stability is very important if a zinc atom is present in the thermal stability due to the “zinc-burning” phenomenon. It can be seen in Figure 4 that there was no “zinc-burning” phenomenon on PVC samples stabilized with pure DMAE-Zn. Furthermore, the PVC sample stabilized by 3 phr of DMAE-Zn + 1 phr of ZnSt_2_ also had good long-term thermal stability. All the results indicate that the ZnCl_2_ produced during the replacement of the active chlorine atom or the neutralization of HCl can be chelated by –OH of DMAE-Zn in situ (illustrated in 
Scheme 2b,c) and generate compounds II and III. In order to verify the complexation of DMAE-Zn with ZnCl_2_, oven thermal aging experiments were conducted to test the thermal stability of PVC samples stabilized by 4 phr of pure ZnCl_2_, and 4 phr of different proportions of ZnCl_2_/DMAE-Zn mixtures. The results are shown in Figure 9. PVC samples containing 4 phr ZnCl_2_ quickly darkened in 10 min. This demonstrated that ZnCl_2_ accelerated the thermal degradation of PVC, meaning that the “zinc-burning” phenomenon occurred. With the increase of the ratio of DMAE-Zn, the initial color and long-term stability of PVC stabilized by ZnCl_2_/DMAE-Zn mixture can be improved. In particular, the PVC samples with 1 phr of ZnCl_2_ + 3 phr of DMAE-Zn exhibited excellent thermal stability, as shown in Figure 9, since the initial color did not change within 50 min and became black completely after being heated for 70 min. In the end, the PVC samples with 1 phr of ZnCl_2_ + 3 phr of DMAE-Zn also showed the “zinc-burning” phenomenon, because it exceeded the ability of DMAE-Zn to chelate ZnCl_2_.

## 4. Conclusions

In this study, DMAE-Zn was synthesized through an alcoholysis reaction. The chemical formations were confirmed through FT-IR. The thermal stability of PVC was studied using thermal aging tests, conductivity tests, UV-VIS spectroscopy tests, and torque rheology tests. The results showed that PVC stabilized by DMAE-Zn had a good initial color and long-term stability, as well as good plasticizing performance. Moreover, the results of the oven thermal aging experiments indicated that there was good synergy among DMAE-Zn, CaSt_2_, and ZnSt_2_ in improving the durability of PVC under long-term exposure to heat. Furthermore, the thermal stabilizing mechanism of DMAE-Zn on PVC was studied through FT-IR, HCl absorption capacity tests, and compound experiments with ZnSt_2_. The results confirmed that DMAE-Zn has the common characteristic of metal alkoxides, which can readily neutralize HCl. FT-IR showed that it was the alkoxy group of DMAE-Zn that replaced the unstable chlorine atoms on PVC chains to improve the initial color and thermal stability of PVC. The HCl absorption capacity of DMAE-Zn (131.4 mg/g) was larger than that of ZnSt_2_ and CaSt_2_, indicating that DMAE-Zn can improve the long-term thermal stability of PVC. It was also found that there were plenty of hydroxy groups on DMAE-Zn which could complex ZnCl_2_ to delay or avoid the “zinc-burning” phenomenon.

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
