# Peer review of "Facile Synthesis of Di-Mannitol Adipate Ester-Based Zinc Metal Alkoxide as a Bi-Functional Additive for Poly(Vinyl Chloride)"

_polymers, 2019, doi:10.3390/polym11050813_

Reviewer 1 Report

I reviewed your intriguing manuscript, which shows the  new synthesis of di-mannitol adipate ester-based  zinc metal alkoxide as bi-functional additives for  PVC. You provide a detailed series of investigations, Fourier transform infrared (FT-IR) spectroscopy and thermogravimetric analysis.

The work has scientific merits, and is generally well written. The experiments are well designed, and most conclusions are supported by the data. However, a range of short comings attracted my attention, which need to be addressed to improve your submission:
1.    Please clearly write what is the purpose of the research

2.        please explain all the abbreviations in the manuscript

3. please describe the substances used in the work more accurately

4. why the aging time was so short

5. please improve the quality of figure 6

6. I do not understand the sense of figure 8

7. which means wording: improving the thermal stability???

Author Response

Response to Reviewer 1 Comments

Reviewer#1

Comments to the Author

I reviewed your intriguing manuscript, which shows the new synthesis of di-mannitol adipate ester-based zinc metal alkoxide as bi-functional additives for PVC. You provide a detailed series of investigations, Fourier transform infrared (FT-IR)  spectroscopy and thermogravimetric analysis.

The work has scientific merits, and is generally well written. The experiments are well designed, and most conclusions are supported by the data. However, a range  of short comings attracted my attention, which need to be addressed to improve your  submission:

Point 1: Please clearly write what is the purpose of the research. 

Response 1: We are very grateful to the Reviewer#1 for pointing out this shortcoming. We had added the purpose of this research in Induction (line 72 to 75): In this study, in order to low the melting point of metal alkoxides, adipic acid was used to synthesize a new kind of thermal stabilizers, di-mannitol adipate ester-based zinc metal alkoxide (DMAE-Zn). The reason why adipic acid was chosen was because it had good compatibility with PVC which made the DMAE-Zn having plasticizing effect on PVC.

The changed part was marked in blue font.

Point 2: Please explain all the abbreviations in the manuscript

Response 2: We are quite sorry for our careless, and thank Reviewer#1 for pointing out the problem. We had check and explain all of the abbreviations.

The changed part was marked in blue font.

Point 3: Please describe the substances used in the work more accurately.

Response 3: We are very appreciated for the Reviewer’s good suggest. We have carefully checked and added detailed descriptions of the substances used in our work (please see 2.1. Materials).

The changed part was marked in blue font.

Point 4: Why the aging time was so short.

Response 4: We are very appreciated for the Reviewer’s comments. It is true that the aging time of PVC stabilized by 4 phr of DMAE-Zn is short which can be seen from Fig. 4. Lead salt stabilizers are the best PVC thermal stabilizers. We have tested the aging time of PVC stabilized by lead salts [Study on pentaerythritol-zinc as a novel thermal stabilizer for rigid poly (vinyl chloride), Journal of Applied Polymer Science, 2012, 126(2): 569-574] and the results (Please see the picture below) show that the color of PVC stabilized by 4 phr of lead salts do not change within 50 min and start to turn into light grey from 70 min, start browning from 110 min, and turn into brownish black at 170 min. Because lead salt is harmful to the human body and the environment, it is gradually banned. At present, calcium/zinc stearic are more widely used as PVC thermal stabilizers. However, the long-term thermal stability of PVC stabilized by Ca/Zn stearic based thermal stabilizers are not good enough which are also proved by our tests. Fig. 4 shows that PVC stabilized by Ca/ZnSt2 starts to turn brown at 20 min and turn black at 40 min quickly. As for DMAE-Zn synthesized in this work, Fig. 4 shows that PVC stabilized by 4 phr of DMAE-Zn starts to turn light brown from 40 min and does not turn into black until 100 min. It proves that, although the aging time of DMAE-Zn is less than lead salts, much bigger than that of Ca/ZnSt2. Especially, Fig. 6 shows that, when DMAE-Zn is compounded with ZnSt2 as PVC thermal stabilizer, there is almost no discoloration within 70 min showing that the aging time of DMAE-Zn@ZnSt2 is quite close to the lead salt stabilizers. Furthermore, DMAE-Zn has good plasticizing performance.

Point 5: Please improve the quality of figure 6.

Response 5: We thank Reviewer#1 for his/her very meticulous observation and for pointing out this error. We have corrected this accordingly. Please refer to the revised Fig. 6 in Page 9 of the revised manuscript.

Point 6: I do not understand the sense of figure 8.

Response 6: We apologize for not explaining the Fig. 8 clearly. The results of the thermal aging tests showed that the addition of DMAE-Zn could improve the initial color and long-term thermal stability of PVC. Moreover, based on the results of our previous quantum chemistry calculations, we found that, due to its high electronegativity, the alkyl oxygen of metal alkoxides had a tendency to attack the carbon atoms (given its high positive charge) attached to allyl chloride in PVC chains [Synergism of pentaerythritol-zinc with b-diketone and calcium stearate in poly(vinyl chloride) thermal stability, Polymer Journal, 2013, 45: 775-782]. At the same time, allyl chloride with a high electronegativity would attack the zinc atom (having high positive charge) of the metal alkoxides. Therefore, the thermal stability mechanism of DMAE-Zn on PVC was proposed that DMAE-Zn could replace the allyl chloride in PVC molecule which was showed with Scheme 2a, and compound I was formed at last. There was -C-O-C- bond in compound I. In order to detect the existence of -C-O-C-, FT-IR was used to test four PVC samples and the results were shown in Fig. 8. Fortunately, there were obvious absorption peaks at 1096 cm-1 (which was the characteristic peaks corresponding to -C-O-C-) in PVC samples stabilized by DMAE-Zn. At the same time, it also confirmed that the proposed thermal stability mechanism of DMAE-Zn on PVC was reasonable.

In order to make it easier to understand Fig. 8, we have added some descriptions about Fig. 8 (please see line 362 to line 366).

The changed part was marked in blue font.

Point 7: Which means wording: improving the thermal stability???

Response 7: We thank the reviewer for the good question and we are sorry for our inaccurate expression. We have revised them carefully.

Page 9 line 268: “DDMA is most efficient in improving the thermal stability of PVC” changed into “DMAE-Zn is the most efficient in preventing PVC molecules from forming conjugated structure, and thus preventing thermal degradation of PVC”.

Page 14 line 401: “improving the thermal stability of PVC” changed into “improving the long-term thermal stability of PVC color”.

The changed part was marked in blue font.

Reviewer 2 Report

The research work done by Degang Li and Lipeng Zhang co-workers “Facile synthesis of di-mannitol adipate ester-based zinc metal alkoxide as bi-functional additives for PVC” is a good research work to study and designed for bi-functional additives for PVC.

Authors documented a new process and the synthesis of di-mannitol adipate ester-based zinc metal alkoxide. The major advantage of the present documented report is the synthesized material can be used as bi-functional additives for PVC.

There are not many literature procedures, similar to present documented report in the literature for new developments as additives for PVC.

Given the importance of practicality for this work, I recommend the publication of this manuscript in the Polymers.

Author Response

Response to Reviewer 2 Comments

Reviewer#2

Point 1: The research work done by Degang Li and Lipeng Zhang co-workers “Facile  synthesis of di-mannitol adipate ester-based zinc metal alkoxide as bi-functional  additives for PVC” is a good research work to study and designed for bi-functional  additives for PVC.  

Authors documented a new process and the synthesis of di-mannitol adipate  ester-based zinc metal alkoxide. The major advantage of the present documented  report is the synthesized material can be used as bi-functional additives for PVC.  

There are not many literature procedures, similar to present documented report  in the literature for new developments as additives for PVC.  

Given the importance of practicality for this work, I recommend the publication of this manuscript in the Polymers.

Response 1: Thanks for your encouraging comments! 

Round  2

Reviewer 1 Report

I accept the manuscrypt in present form